# miRNA Expression Profiling in Subcutaneous Adipose Tissue of Monozygotic Twins Discordant for HIV Infection: Validation of Differentially Expressed miRNA and Bioinformatic Analysis

**DOI:** 10.3390/ijms23073486

**Published:** 2022-03-23

**Authors:** Elena Bresciani, Nicola Squillace, Valentina Orsini, Roberta Piolini, Laura Rizzi, Laura Molteni, Ramona Meanti, Alessandro Soria, Giuseppe Lapadula, Alessandra Bandera, Andrea Gori, Paolo Bonfanti, Robert John Omeljaniuk, Vittorio Locatelli, Antonio Torsello

**Affiliations:** 1School of Medicine and Surgery, University of Milano-Bicocca, 20900 Monza, Italy; laura.rizzi@unimib.it (L.R.); laura.molteni@unimib.it (L.M.); ramona.meanti@unimib.it (R.M.); giuseppe.lapadula@unimib.it (G.L.); paolo.bonfanti@unimib.it (P.B.); vittorio.locatelli@unimib.it (V.L.); antonio.torsello@unimib.it (A.T.); 2Infectious Diseases Unit, Azienda Socio Sanitaria Territoriale di Monza, San Gerardo Hospital, 20900 Monza, Italy; n.squillace@asst-monza.it (N.S.); valentina.orsini@asst-monza.it (V.O.); alessandro.soria@asst-monza.it (A.S.); 3Department of Clinical Sciences L. Sacco, Infectious and Tropical Diseases Section, University of Milano, 20157 Milano, Italy; roberta.piolini@unimi.it; 4Clinic of Infectious Diseases, IRCCS Ca’ Granda Ospedale Maggiore Policlinico di Milano Foundation, 20122 Milan, Italy; alessandra.bandera@unimi.it (A.B.); andrea.gori@unimi.it (A.G.); 5Department of Pathophysiology and Transplantation, University of Milan, 20122 Milan, Italy; 6Department of Biology, Lakehead University, Thunder Bay, ON P7B 5E1, Canada; romeljan@lakeheadu.ca

**Keywords:** combined highly antiretroviral therapy (cART), HALS (HIV-associate lypodistrophy syndrome), miRNA, metabolic disorders

## Abstract

Combined AntiRetroviral Treatments (cARTs) used for HIV infection may result in varied metabolic complications, which in some cases, may be related to patient genetic factors, particularly microRNAs. The use of monozygotic twins, differing only for HIV infection, presents a unique and powerful model for the controlled analysis of potential alterations of miRNAs regulation consequent to cART treatment. Profiling of 2578 mature miRNA in the subcutaneous (SC) adipose tissue and plasma of monozygotic twins was investigated by the GeneChip^®^ miRNA 4.1 array. Real-time PCR and ddPCR experiments were performed in order to validate differentially expressed miRNAs. Target genes of deregulated miRNAs were predicted by the miRDB database (prediction score > 70) and enrichment analysis was carried out with g:Profiler. Processes in SC adipose tissue most greatly affected by miRNA up-regulation included (i) macromolecular metabolic processes, (ii) regulation of neurogenesis, and (iii) protein phosphorylation. Furthermore, KEGG analysis revealed miRNA up-regulation involvement in (i) insulin signaling pathways, (ii) neurotrophin signaling pathways, and (iii) pancreatic cancer. By contrast, miRNA up-regulation in plasma was involved in (i) melanoma, (ii) p53 signaling pathways, and (iii) focal adhesion. Our findings suggest a mechanism that may increase the predisposition of HIV^+^ patients to insulin resistance and cancer.

## 1. Introduction

HIV-associated lipodystrophy syndrome (HALS) is a phenotypically heterogeneous and acquired condition [1] characterized by a redistribution of adipose tissue, such as peripheral lipoatrophy of subcutaneous adipose tissue (SC) (in the face, limbs, and buttocks) and/or lipohypertrophy in the visceral depots and lipomatosis in the dorsocervical area (buffalo hump) [2]. HALS is often associated with systemic metabolic disorders, including dyslipidemia and insulin resistance, which expose the patients to an enhanced risk of developing cardiovascular diseases and type II diabetes [2]. HALS pathogenesis is multi-factorial and not yet completely understood [3]. Recognized predisposing factors for HALS include (i) HIV infection, (ii) combined highly antiretroviral therapy (cART), (iii) gender, (iv) an elevated T-CD4^+^ lymphocyte count, (v) a baseline viral load, (vi) duration of the antiretroviral therapy, (vii) bodyweight before starting treatment, and (viii) genetic background [4,5].

Genome-wide association studies (GWAS) have identified single nucleotide polymorphisms (SNPs) related to metabolic changes useful for predicting an individual’s susceptibility to developing the pathology and the response to therapy [6]. In particular, SNPs in the gene of *Insulin Receptor Substrate 1* (IRS-1), *low-density lipoprotein receptor* (LDLR), and *apolipoprotein E* (APOE), were recognized to influence the risk of body fat changes in HIV patients on cART [5].

In addition to SNIPs, has been proposed a microRNAs (miRNAs) involvement in the development of HALS. MiRNAs are small non-coding RNAs (about 22 nucleotides in length) that regulate gene expression at the post transcriptional level by interacting with the 3′-UTR region of target mRNAs. MiRNAs are involved in several cellular processes, and modifications in their expression have been associated with many human diseases [7]. MiRNAs included in exosomes [8] from many sources such as the adipose tissue [9,10] are released into bio-fluids mediating cell-cell communication [7]. Circulating miRNAs are considered possible biomarkers for different pathologies [10], including metabolic diseases like insulin resistance [11] and type II diabetes mellitus [3,12,13].

In particular, in HIV lipodystrophy patients it has shown an altered pattern of circulating miRNAs [14]. Given the emerging relevance of miRNAs in several pathological conditions, we investigated miRNA expression profiles in SC and plasma samples of two subjects. The peculiarity of this study is to consider a pair of monozygotic twins, of which one is HIV^+^ on cART and affected by moderate/severe facial lipoatrophy and the other one is HIV^−^, without co-morbidities. The unicity of this condition allows us to study specific miRNAs engaged in the development of metabolic and morphological features distinctive of HALS, independent of genetic background, to identify potential biomarkers with a diagnostic and/or prognostic value in respect of HALS.

## 2. Results

### 2.1. Clinical Data

The clinical data of the twins are summarized in Table 1. Significant differences were observed between HIV^+^ and HIV^−^ subjects only regarding BMI and waist circumference parameters (the BMIs for the HIV^+^ and HIV^−^ twins were 21.8 and 27.3, respectively). The HIV^−^ individual presented a BMI value corresponding to the overweight condition and an abdominal circumference value slightly higher than that corresponding to low cardiovascular risk.

### 2.2. Differentially Expressed microRNA (DEGs) in the SC Adipose Tissue and the Plasma of the Twins

A total of 2578 human mature miRNAs in SC adipose tissue and plasma were profiled through the microarray technique. Of the mature miRNAs examined in SC and plasma, which showed a fold-change of >+2 to <−2 in the HIV^+^ twin compared to the HIV^−^ twin, 95 were observed in SC, and 198 in the plasma; these values represent 3.7% and 7.7%, respectively, of the total differentially expressed miRNAs, DEGs.

In SC adipose tissue, 30 miRNAs were up-regulated, whereas the other 65 were down-regulated. In the plasma, by comparison, 118 were up-regulated, whereas 80 were down-regulated. Notably, miR-642b-3p was down-regulated in SC adipose tissue as well as in plasma. By contrast, miR-3620-5p, miR-6779-5p, and miR-6803-5p were down-regulated in SC adipose tissue but up-regulated in plasma. The heatmaps show the hierarchical clustering of DEGs in SC adipose tissue (Figure 1A) and plasma (Figure 1B). The Venn diagram shows the number of shared DEGs between SC and plasma (Figure 1C).

### 2.3. Validation of microRNA DEGs in SC Adipose Tissue by Real-Time qPCR

DEGs in SC adipose tissue analyzed through Real-Time qPCR were chosen according to their expression values: miR-15b-5p, miR-532-3p, and miR-127-3p were selected among the up-regulated miRNAs, while miR-122-5p, miR-6803-5p, and miR-3620-5p were among the down-regulated ones. Table 2 summarizes the expression and the FC values for each miRNA obtained from microarray and Real-Time qPCR analysis.

The microarray expression profiles of miR-15b-5p and miR-127-3p were confirmed by Real-Time qPCR, using both the housekeeping genes. Significantly, miR-532-3p was resulted up-regulated in the HIV^+^ SC adipose tissue sample only when the expression values are normalized for U6, (Figure 2). Of the down-regulated miRNAs, Real-Time qPCR analysis revealed that miR-122-5p and miR-3620-5p had expression profiles consistent with the microarray results when U6 and miR-425-5p were employed as reference genes. MiR-6803-5p, however, was validated only when normalized with respect to miR-425-5p (Figure 3).

### 2.4. Analysis of microRNA DEGs in Plasma by Real-Time qPCR

Among the miRNA DEGs in the plasma obtained by the microarray analysis, the most significant in terms of over-expression (miR-4529-3p, miR-16-5p, miR-6803-5p, and miR-668-5p) and under-expression (miR-877-5p and miR-642b-3p) were selected for validation by Real-Time qPCR (Table 3). Let-7i-5p and miR 425-5p were used as reference genes.

However, the analysis in this case only partially confirmed the microarray experiment outcome; the miR-6803-5p expression profile agreed with the microarray data only when the expression values were normalized for both housekeeping genes (Figure 4). The results from other selected miRNAs were indeterminate.

### 2.5. Analysis of microRNA DEGs in Plasma by ddPCR

One disadvantage of the Real-Time qPCR technique for plasma samples is a low sensitivity; consequently, the absolute quantification of their miRNAs was evaluated by ddPCR. According to the microarray data, miR-16-5p (Figure 4A), miR-6803-5p, miR-668-5p, and miR-4529-3p were up-regulated in HIV^+^ plasma in comparison to HIV^−^ plasma (in direct comparisons of HIV^+^ vs. HIV^−^ plasma respectively, 2306 miRNA copies/µL vs. 1113 copies/µL; 22 miRNA copies/µL vs. 4 miRNA copies/µL; 17 miRNA copies/µL vs. 12 miRNA copies/µL; and 26 miRNA copies/µL vs. 16 miRNA copies/µL) (Figure 4B). The miR-4707-5p levels, as well as those of selected down-regulated miRNAs (i.e., miR-877-5p, miR-642b-3p), were not measurable even with the ddPCR technology.

### 2.6. Analysis of Predicted Target Genes and Functional Annotations of DEGs microRNAs

In order to explore potential roles of the validated miRNAs in SC adipose tissue (miR-15b-5p, miR 532-3p, miR-127-3p, miR-122-5p, miR-6803-5p, and miR-3620-5p) and plasma (miR-16-5p, miR-4529-3p, miR-6803-5p, miR-877-5p, and miR-642b-3p) in the context of predicted target genes, a bioinformatics analysis was conducted using miRDB, a conservative approach. The putative target genes were functionally characterized by performing GO (version released 2019-06-01) and KEGG pathway enrichment analysis (release 90.0, 2019-04), which was carried out on up-regulated and down-regulated miRNAs for both types of samples, employing g: Profiler. It uses the GO and the Kyoto Encyclopedia of Genes and Genomes (KEGG) databases to identify significant functional biological annotations (biological processes, cellular component, and molecular functions), as well as pathways related to the target genes, sorting them by statistical significance.

The most represented categories among the biological processes of the up-regulated miRNAs in SC adipose tissue were those linked to macromolecule metabolic process regulation of neurogenesis, and protein phosphorylation (Figure 5A). KEGG analysis revealed their involvement in insulin signaling pathways, neurotrophin signaling pathways, and pancreatic cancer (Figure 5B). Plasma up-regulated miRNAs, by comparison, were implicated in melanoma, p53 signaling pathways, and focal adhesion (Figure 5C).

### 2.7. Analysis of Lipin-1 and miRNA-218-5p Expression in SC Adipose Tissue

The present findings are consistent with previous results which suggest an inverse relationship between miRNA 218-5p and lipin-1 expression, one of its target genes [15,16]. A similar pattern of expression was observed in these twins; specifically, miRNA 218 5p was up-regulated and lipin-1 was down-regulated in the HIV^+^ twin, compared to the HIV^−^ twin (Figure 6).

## 3. Discussion

The underlying mechanisms of HIV-associated lipodystrophy syndrome (HALS) are not completely understood, and research in this field is still ongoing. This pathology is consequent to multiple contributing and interacting factors, including (i) the HIV infection, (ii) multiple pharmacological treatments, and (iii) host factors, such as genetic background. In this regard, it has been reported that specific polymorphisms might affect the risk of body fat changes in HIV^+^ subjects receiving combined antiretroviral therapy (cART) [17,18,19], whereas other elements contribute to the development of dyslipidemia and insulin resistance [20,21,22] The clinical case that we report in this study, consisting of a pair of monozygotic twins, of which one on cART treatment was affected by HALS, is very rare. Indeed, the individuals were discordant only for their pathology and pharmacological treatment; this aspect is relevant as it could allow the identification of metabolic and molecular alterations not linked to the genetic components.

Recent studies have focused on the impact of miRNAs in the regulation of many cellular functions and gene networks in different pathologies. As a consequence of their high stability in various tissues and body fluids, miRNAs may be considered potential biomarkers [10]. These miRNAs have been demonstrated to be important regulators of the development and physiological state of metabolically active tissues; dysregulation of their expression can result in impaired glucose and lipid homeostasis, suggesting involvement in metabolic diseases [23]. A potential role of miRNAs has been postulated also in the onset of HALS. Mice knock-out (KO) for Dicer, a ribonuclease involved in miRNA biogenesis, display a form of lipodystrophy reminiscent of the human HALS, including the loss of subcutaneous (SC) fat, insulin resistance, and dyslipidemia [14]. Dicer mRNA expression appears also downregulated in SC fat depots (i.e., dorsocervical and abdominal) of HIV^+^ patients [14,15,16,17,18,19,20,21,22,23,24], suggesting that its reduced expression could lead to a broad altered expression profile of several cellular miRNAs [25]. Accordingly, in samples of SC adipose tissue of lipodystrophic HIV^+^ patients, the upregulation of specific miRNAs, which controls the expression of genes involved in adipocyte differentiation and apoptosis was observed [15]. Among these, miR 218-5p inhibits lipin-1 mRNA expression, as this appeared decreased in 3T3-L1 cell lines following protease inhibitors (PIs) treatment, while miR-218-5p expression was enhanced and resulted up-regulated when the cells were treated with a specific miR-218-5p inhibitor [16].

In our research, microarray analysis identified the miRNAs differentially expressed (DEGs) between the twins: 95 in SC adipose tissue and 198 in plasma. To consolidate the outcome of the microarray, selected miRNAs were thereafter validated by Real-Time qPCR, as microarray technology is prone to create false positive or false negative results [26]. Real-Time qPCR is a widely used method to quantify differences in the expression profile of genes because of its high sensitivity, specificity, and reproducibility. However, an important aspect to consider is the normalization process required to reduce the analytical variability and avoid potential data misinterpretation [27]. In particular, as regards the miRNAs expression analysis, the choice of the housekeeping genes constitutes one of the most critical issues, as there are no reference genes univocally accepted for this purpose, especially for circulating miRNAs [27,28]. One of the criteria generally adopted, as well as in the current case, is the a priori selection of the reference genes, according to the literature. However, to confer more strength to the outcome, we combined this approach with a specific evaluation of the experimental condition examined. In particular, miR-425-5p was identified through microarray expression values analysis, indicating stable levels between the SC adipose tissue and plasma samples of the twins. Accordingly, other studies have described the use of miR-425-5p as a reference gene, in combination with other two miRNAs in a hepatitis B-virus cell model [29]. The same assessment was made also for the other chosen reference genes. U6 is one of the most commonly employed, although its use is a matter of debate; it is not properly a miRNA [28,30]. Some studies have indicated its unsuitability for miRNA normalization due to expression instability in various experimental conditions [31,32,33]. Conversely, in other models, U6 expression levels are very stable among the sample groups, making it a valid endogenous control [30,34]. Concerning the miRNAs present in the plasma, few reference data are available in support of selecting a reliable endogenous reference gene. Let-7i is part of miRNAs combinations and is generally used to normalize circulating miRNAs expression in different types of investigational scenarios [35,36].

Although there were some minimal discrepancies between microarray and Real-time qPCR data in the SC samples, due mainly to reference genes used in the normalization, the principal difficulty was the lack of detectable levels of miRNAs in the plasma by Real-Time qPCR. To overcome this issue, we used the ddPCR, a more sensitive and innovative technology able to detect very low concentrations of viral RNA or DNA [37,38] and circulating miRNAs [39,40,41,42] performing an absolute quantification of target nucleic acids and avoiding data normalization. Our ddPCR results indicate a high concentration for miR-16-5p and miR-6803-5p in the HIV^+^ plasma sample compared to HIV^−^, in agreement with microarray data.

To investigate the meaning of miRNAs expression alteration in the samples more meaningfully, we adopted a bioinformatics conservative approach in an effort to identify the protein targets and find pathways controlled by the miRNAs under consideration. Pathway enrichment analysis showed that some of the examined miRNAs were associated with the insulin signaling pathway. Indeed, the liver-specific miR-122-5p is linked to insulin biosynthesis [43] but also plays a role in hepatic cholesterol and lipid homeostasis [44]. Among miR-122-5p target genes, is the transcript codifying for the *diacylglycerol acyl-CoA acyltransferase* (DGAT1), which converts diacylglycerol (DAG) into triacylglycerol (TG). Hence, it may be speculated that the miR-122-5p down-regulation observed in the HIV^+^ SC adipose tissue may increase the expression of DGAT1 as well as TG synthesis, which ultimately may result in hypertriglyceridemia, one of the conditions affecting HALS patients [2]. The miR-127-3p was over-expressed in the SC adipose tissue sample of the HIV^+^ twin; elevated miR-127-3p levels have also been found in pancreatic islets, particularly in islets derived from glucose-intolerant donors compared to those from healthy donors. In control subjects miR-127-3p expression in pancreatic islets positively correlates with insulin mRNA expression and negatively to glucose-stimulated insulin secretion, suggesting its involvement in the physiological regulation of insulin biosynthesis and secretion. The lack of a comparable pattern in pancreatic islets from glucose intolerance patients may be evidence of an impaired miRNA network in insulin function of diabetic patients [43]. Considering these findings, although our analysis was conducted in a different area of the body, we speculate that an alteration of miR-127-3p could interfere with the insulin pathway, for instance in SC adipose tissue, ultimately causing insulin resistance, a typical mark of the HALS.

Lipoatrophy is the HALS main clinical feature displayed by the twin HIV^+^. A link between lipoatrophic SC adipose tissue of HALS patients and the decreased expression of the adipogenic transcription factors as well as of the genes involved in adipocytes’ differentiation has been described [45]. Among the up-regulated miRNAs identified in the SC adipose tissue of the HIV^+^ twin, the miR-15b-5p and 532-3p share as target genes the *transcription factor HMGA2 (high-mobility group AT-hook 2)*. HMGA2 promotes adipocyte differentiation, as it is required for the expression of PPARγ [46]. Since both miRNAs are upregulated in the SC adipose tissue sample of HIV^+^ twin, HMGA2 expression may be inhibited, thereby impairing adipogenesis. Likewise, blocking the expression of lipin-1, which is essential for adipocyte differentiation, leads to the inhibition of the adipogenic process [47]. Accordingly, in our case, lipin-1 is down-regulated in the SC adipose tissue sample of the HIV^+^ subject, and in keeping with previous reports [15,16], lipin-1 mRNA levels are inversely correlated to miR-218-5p expression, providing further insight into the mechanisms causing lipoatrophy.

Research focused on the identification of circulating miRNAs as biomarkers has been accelerated in recent years because of its potential diagnostic or prognostic value in different pathological contexts as well as the possibility of monitoring pathology progression and predicting a therapeutic outcome [48,49,50,51,52]. Furthermore, since miRNAs can be transported in plasma by high-density lipoprotein (HDL) or exosomes [8,53], it is assumed that plasma miRNAs may be delivered to the recipient cells, where they can regulate target gene expression at a distance. Concerning circulating miRNAs examined, very few data are available, except for the miRNA miR-16-5p, which was up-regulated in the HIV^+^ twin compared to the HIV^−^ twin. This has been identified in miRNAs signatures in different cancer conditions [49,52]. In this regard, miR-16-5p belongs to the miR-15a/16 gene cluster, and several studies showed their role in the induction of cancer cell apoptosis and inhibition of proliferation, angiogenesis, invasion, and metastasis processes, demonstrating their tumor-suppression activities [54]. In addition, a positive correlation between miR-16-5p up-regulation and HIV infection has been detected in Kaposi Sarcoma samples [55].

Although the very particular case taken into consideration in our research allowed us to compare two individuals identical from the genomic point of view, providing thus the unique opportunity to overcome the inter-individual differences, it presents a limitation due to the low sample size. Another weakness could regard the conservative approach adopted to perform the bioinformatics analysis. Although we are aware that computational conservative methods used in target prediction generally only identify about 80% of known bindings, excluding non-canonical targets, we decrease the probability of false positives and negatives [56].

However, the results presented provide an overview of the miRNAs signature and might represent a challenge to validate further the abundance levels of the identified miRNAs in a larger and more representative cohort of monozygotic twins.

In conclusion, we identified a set of deregulated miRNAs in the monozygotic HIV-affected twin compared to his non-HIV twin. The upregulation of certain miRNAs linked to insulin signaling and metabolic factors involved in adipogenic processes strengthens the idea of its contribution to lipid homeostasis. Conversely, miRNA deregulation could also explain the higher prevalence of cancer in HIV^+^ subjects that is only partially explained by immunodeficiency. MiRNAs studies could improve our knowledge of mechanisms that produce metabolic derangements in HIV patients.

## 4. Materials and Methods

### 4.1. Participants

A pair of monozygotic twins (age 49 years) who participated in this study were denoted (i) Twin A (HIV^+^, with generalized moderate/severe lipoatrophy and on antiretroviral treatment with tenofovir diproxil fumarate, emtricitabine, and efavirenz since 1998, and (ii) Twin B (HIV^−^, without co-morbidities). Pre-surgical physical assessments included measurements of (i) BMI, (ii) waist size, and (iii) systolic and diastolic blood pressure. They also included plasma levels of (iv) glucose, (v) ALT, (vi) HDL, (vii) total cholesterol, and (viii) triglycerides. This study was approved by the San Gerardo Hospital Ethical Committee (approval # 1757), and written informed consent was obtained from the subjects prior to commencing the study.

### 4.2. microRNA Extraction from SC Adipose Tissue and Plasma Samples

SC adipose tissue samples were obtained by a dysplastic mole removal procedure, or by liposuction surgery from the abdominal region, from the HIV^+^ twin and the HIV- twin, respectively. Tissues were washed immediately with sterile physiological saline, then stored in RNA later^®^ (Ambion) at −20 °C until a subsequent RNA extraction assay. Immediately prior to surgery, plasma samples were obtained and collected into EDTA-treated tubes and immediately frozen at −80 °C. Total RNA was extracted from SC adipose tissue and plasma samples accordingly to manufacturer’s instructions, using miRCURY™ RNA Isolation Kit-Cell and Plant (Exiqon) and miRCURY™ RNA isolation kit-Biofluids (Exiqon).

### 4.3. microRNA Microarray

The Affymetrix GeneChip^®^ miRNA 4.1 array strip (which contains 2578 human mature miRNAs probes) was used for global miRNA expression analysis. An aliquot of RNA (1 μg for SC adipose tissues and 8 μL for plasma samples) was labeled using the FlashTag™ Biotin RNA Labeling Kit (Applied Biosystem), followed by hybridization overnight according to the manufacturer’s instructions. Afterward, the miRNA microarray strips were stained and washed using GeneChip Fluidics Station 450 (Affymetrix). The images obtained from the microarrays were scanned by Affymetrix GeneChip Scanner 3000 (Affymetrix).

### 4.4. microRNA Data Analysis

The quality control of the scanned data was first estimated by confirming the order of the signal intensities of the Poly-A and Hybridization controls using an Affymetrix GeneChip Expression Console (Affymetrix). The raw expression values were imported as Affymetrix.CEL files into a Partek Genomics Suite 6.6 (Partek Inc., St. Louis, MO, USA). The data were analyzed and normalized including the Preprocessing, Differentially Expressed Genes (DEGs) Finding, and Clustering modules. The RMA algorithm was used to eliminate all signals related to non-specific hybridization and to normalize the microarray data. The mean fluorescence intensities of all genes were obtained comparing two samples (one vs. one) for each “sample type” condition; the analysis was performed comparing the HIV^+^ twin to the HIV^−^ twin. After normalization, the DEGs satisfying the conditions of the Fold Change (FC) cut-off of 2 (>+2 or <−2) from all the genes probed in the strip were selected. Hierarchical cluster analysis was also performed to see how data aggregated and to generate heat maps. For DEGs selection, only the FC method was used as there are only two samples, one for type and it was not possible to apply the statistical analysis procedure.

### 4.5. microRNA Target Prediction and Functional Analysis

The target genes of selected miRNAs were predicted using miRDB-MicroRNA Target Prediction and Functional Study Database v6.0 software (www.mirdb.org, accessed on 26 September 2018), which utilizes a MiRTarget algorithm that evaluates the pairing between the miRNA seed sequence and the target 3′-UTR region. It assigned to each gene a score between 50 and 100. Among the putative target genes of each miRNA, those with a prediction score ≥70 were considered and subsequently analyzed with g:Profiler (https://biit.cs.ut.ee/gprofiler/gost; version e95_eg42_p13_f6e58b9, accessed on 26 September 2018).

### 4.6. Validation of microRNA Microarray Expression Data by Real-Time qPCR and Droplet Digital PCR (ddPCR)

Ten ng of total RNA (adipose tissue) and 2 µL of RNA eluate (plasma) were reverse transcribed in a 10 μL total reaction volume using the Universal cDNA synthesis kit II, as part of the miRCURY LNATM Universal RT microRNA PCR System (Exiqon), per the manufacturer’s instructions. A synthetic UniSp6 RNA spike-in was added to the reaction mix, to provide a control for the quality of the RNA isolation and cDNA synthesis reaction. The expressions of selected miRNAs identified by the microarray study were verified by Real-Time qPCR in a 7900 fast real-time PCR system (Applied Biosystems, ThermoFisher Scientific, Waltham, MA, USA) for the SC adipose tissue, and by ddPCR (Biorad) for the plasma. Three μL of SC adipose tissue cDNA (diluted 1:10) and 3 μL (undiluted) of plasma samples cDNA were amplified using a specific miRCURY PCR primer set (Exiqon) (Table 4). Each sample was conducted in triplicate. The Real-Time qPCR data were analyzed with the ΔΔCT method. The expression of every single miRNA was normalized using hsa-snU6 and hsa-miR-425-5p, and hsa-Let-7i-5p and hsa-miR-425-5p as reference genes, respectively for adipose tissue and plasma (see Table 4). For plasma ddPCR reaction, 3.3 μL of cDNA sample (diluted 1:2, 1:5, and 1:10 accordingly to the microarray expression study) were amplified using QX200 EvaGreen ddPCR Supermix (Biorad) and nuclease-free water up to 22 μL. A total of 20 μL of ddPCR mix reaction and 70 μL of QX200 droplet generation oil for EvaGreen (Biorad) were loaded into a DG8TM Cartridge (Biorad), for droplet generation in a QX200™ Automatic Droplet Generator (Biorad). The droplets were transferred to a 96-well plate (Eppendorf), then heat-sealed using Biorad’s PX1TM PCR Plate sealer and pierceable foil (Biorad). The droplets were amplified by standard PCR using a C100 TouchTM Thermal Cycler (Biorad), with a standard cycling protocol (95 °C for 5 min; 40 cycles of 95 °C for 30 secs, 58 °C for 1 min; 4 °C for 5 min, and 90 °C for 5 min; hold at 4 °C). These were read in the QX200™ Droplet Reader (Biorad) and analyzed by Quantasoft™ version 1.7.4 software (Bio-Rad). The analysis output corresponds to a concentration value (miRNA copies/µL of DNA target in the Reverse Transcription [RT] reaction) for each sample, adjusted for the RT dilution factor.

### 4.7. Analysis of lipin-1 and miRNA-218-5p Expression in SC Adipose Tissue Samples by Real-Time qPCR

RT reaction was performed using either iScriptTM cDNA Synthesis Kit (BioRad) for lipin-1 mRNA expression or TaqMan MicroRNA Reverse Transcription kit (Applied Biosystems, ThermoFisher Scientific) for hsa-miR-218-5p expression, per the manufacturer’s instructions. Real-Time qPCR was performed using Taqman probes (Mm00550511_m1 for lipin-1 and assay 000521 for hsa-miRNA-218) and iTaq Universal Probe Supermix (Bio-Rad) in a 7900HT Fast Real-Time PCR System (Applied Biosystems, ThermoFisher Scientific). The real-time PCR data were analyzed with ΔΔCT method; 18S and snU6 were used to normalize respectively the expression of lipin-1 and has-miR-218-5p.

### 4.8. Statistical Analysis

Given the inability to perform a proper statistical analysis, due to the small sample size (i.e., only two individuals), each sample was run in three to five technical replicates of Real-Time qPCR experiments. For each miRNA, the resulting fold change values were averaged and expressed as mean ± standard error of the mean (SEM). These were analyzed statistically applying the unpaired t-test and employing GraphPad Prism 8.0 (GraphPad Software, Inc., San Diego, CA, USA). A p-value of less than 0.05 was considered significant.

## Figures and Tables

**Figure 1 ijms-23-03486-f001:**
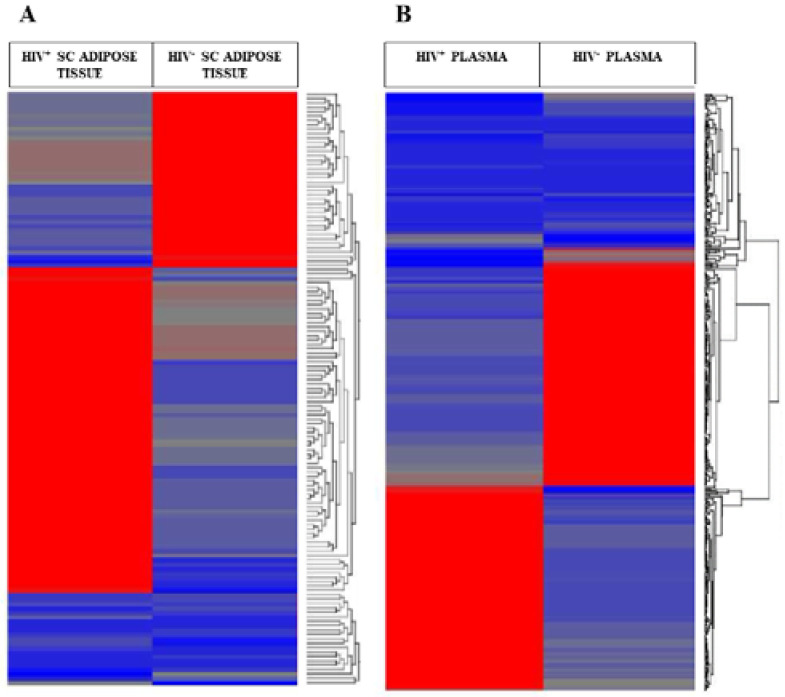
Hierarchical clustering and heatmaps of the miRNA microarray analysis performed on SC adipose tissue (**A**) and plasma (**B**) samples comparing the HIV^+^ twin vs. the HIV^−^ one. Samples names are indicated at the top. The DEGs were selected based on fold change (FC) values >+2 or <−2. Red colors indicate upregulated, and blue colors downregulated probes. (**C**) Venn diagram showing DEGs differentially expressed between the SC and plasma.

**Figure 2 ijms-23-03486-f002:**
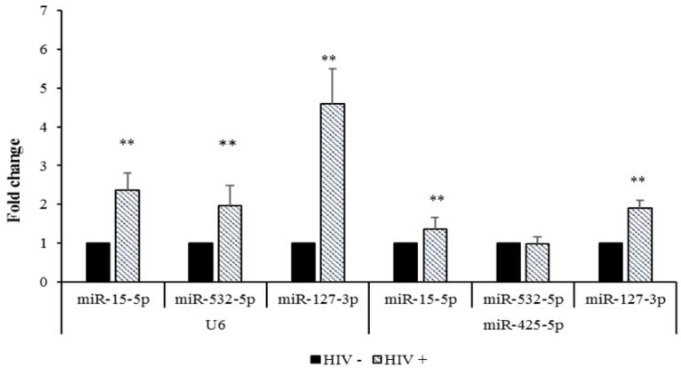
Comparison of the FC values obtained by Real-Time qPCR analysis of upregulated miRNAs in the SC adipose tissue samples, using U6 or miR-425-5p, as reference genes (HIV^−^ twin vs. HIV^+^ one). The FC values (expressed as means ± SEM) were obtained from three to five independent technical replicates of Real-Time qPCR experiments; ** *p* < 0.01.

**Figure 3 ijms-23-03486-f003:**
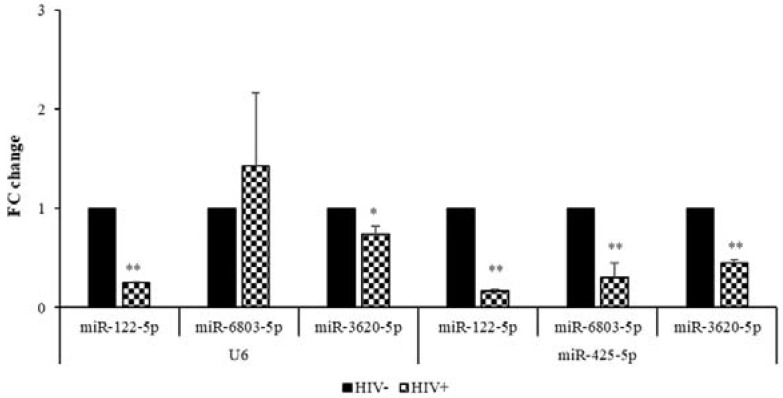
Comparison of the FC values obtained by Real-Time qPCR analysis of downregulated miRNAs in the SC adipose tissue samples, using U6 or miR-425-5p, as reference genes (HIV^–^ twin vs. HIV^+^ one). The FC values (expressed as means ± SEM) were obtained from three to five independent technical replicates of Real-Time qPCR experiments. * *p* < 0.05 and ** *p* < 0.01.

**Figure 4 ijms-23-03486-f004:**
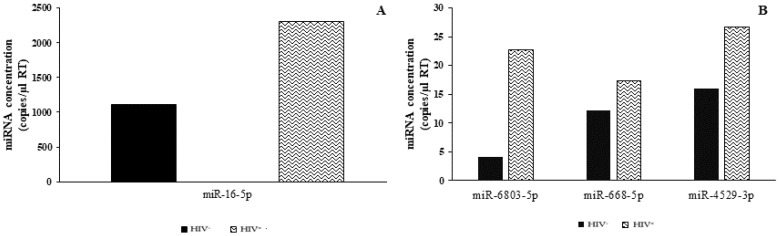
Absolute quantification of miR-16-5p (**A**) and miR-6803-5p, miR-668-5p, and miR-4529-3p (**B**) by ddPCR in the HIV^−^ and HIV^+^ plasma samples. The miRNA concentration is expressed as numbers of copies of DNA target in 10 µL of RT reaction and adjusted for the RT dilution factor.

**Figure 5 ijms-23-03486-f005:**
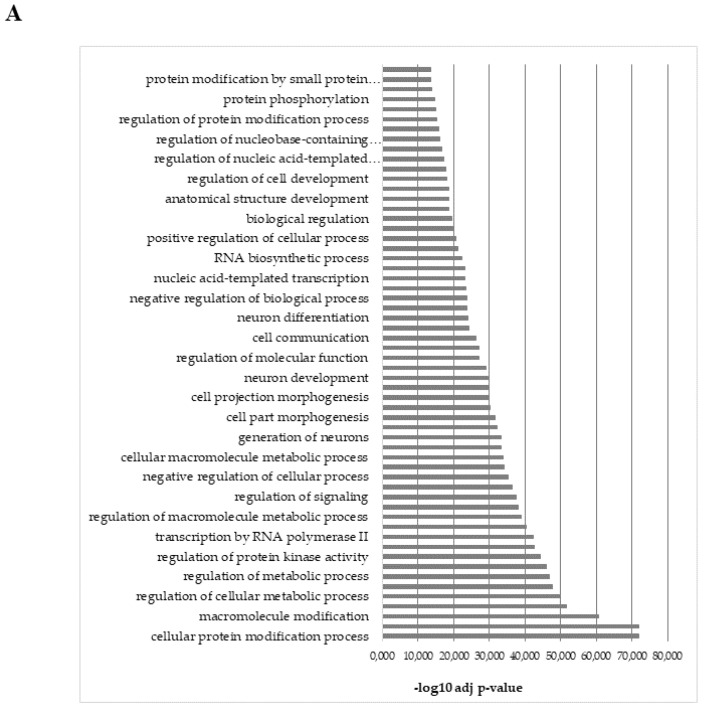
(**A**) Biological processes identified by GO functional enrichment analysis of the miRNAs upregulated in the SC adipose tissue of the HIV^+^ twin; (**B**) Biosynthetic pathways identified by KEGG functional analysis of the predicted target genes controlled by the miRNAs upregulated in the SC adipose tissue of the HIV^+^ twin; (**C**) Biosynthetic pathways identified by KEGG functional analysis of the predicted target genes controlled by the miRNAs upregulated in the plasma. The *p*-value indicates the significance of the pathways correlated to the miRNAs: the lower the *p*-value, the more significant the pathway is. −log10(*p*-value) > 1.30 corresponds to *p* < 0.05.

**Figure 6 ijms-23-03486-f006:**
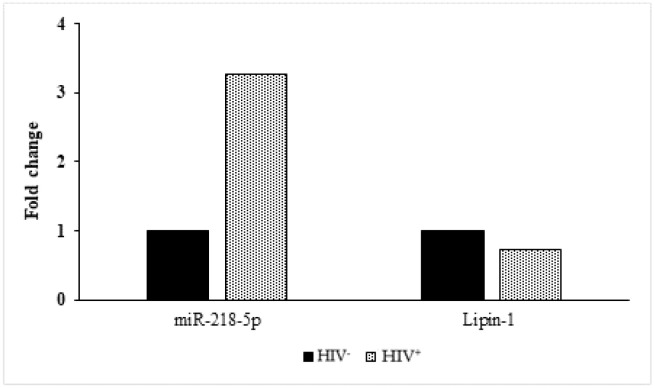
FC values of mRNA expression of miR-218-5p and lipin-1 expression in the HIV^+^ and HIV^−^ twin.

**Table 1 ijms-23-03486-t001:** Clinical data of the HIV^+^ and HIV^−^ twins.

Variables	HIV^+^	HIV^−^
BMI	21.8	27.3
Waist circumference (cm)	80	93
SBP (mmHg)	120	120
DBP (mmHg)	60	80
Glucose (mg/dL)	92	94
ALT (U/L)	32	15
Total cholesterol (mg/dL)	179	172
HDL-Cholesterol (mg/dL)	41	46
Triglycerides (mg/dL)	78	124

**Table 2 ijms-23-03486-t002:** Expression and FC values of upregulated and downregulated miRNAs in the SC adipose tissue samples of the HIV^+^ twin with respect to HIV^−^ one, analyzed by microarray and Real-Time qPCR experiments. In Real-Time qPCR experiments, FC (expressed as means ± SEM) were obtained from three to five independent technical replicates. The normalization was conducted using U6 and miR-425-5p as reference genes.

				Real-Time qPCR FC Means ± SEM	
	miRNAs Expressed in SC ADIPOSE TISSUE	Microarray Expression Values (HIV ^+^ vs. HIV ^−^)	Microarray FC (>+2 or <−2)	Reference Gene U6	Reference Gene miR-425-5p
Upregulated	miR-15b-5p	215/89	2.4	2.36 ± 0.44	1.36 ± 0.29
	miR-532-3p	118/77	2.1	1.96 ± 0.53	0.98 ± 0.17
	miR-127-3p	76/32.8	2.3	4.59 ± 0.92	1.90 ± 0.20
Downregulated	miR-122-5p	637.8/2150	−3.4	0.25 ± 0.01	0.16 ± 0.02
	miR-6803-5p	275.8/694.4	−2.5	1.42 ± 0.74	0.30 ± 0.15
	miR-3620-5p	121.3/263.9	−2.2	0.74 ± 0.08	0.45 ± 0.04

**Table 3 ijms-23-03486-t003:** Expression and FC values of upregulated and downregulated miRNAs in the plasma samples of the HIV^+^ and HIV^−^ twins analyzed by microarray and Real-Time qPCR experiments.

				Real-Time qPCR FC Means	
	miRNAs Expressed in PLASMA	Microarray Expression Values (HIV ^+^ vs. HIV ^−^)	Microarray FC	Reference Gene Let-7i	Reference Gene miR-425-5p
Upregulated	miR-4529-3p	950.5/399.7	2.4	Indetermined	Indetermined
	miR-16-5p	857.2/118	7.3	0.90	0.92
	miR- 6803-5p	333.8/52	6.4	3.91	4.12
	miR-4707-5p	262.8/71	3.1	Indetermined	Indetermined
	miR-668-5p	336.8/156.4	2.2	Indetermined	Indetermined
Downregulated	miR-877-5p	82/244.5	−3.0	2.32	1.89
	miR-642b-3p	1.8/6.0	−3.4	Indetermined	Indetermined

**Table 4 ijms-23-03486-t004:** Primer set used in the Real-Time qPCR validation experiments for miRNAs resulted in up- or down-regulated in SC adipose tissue and plasma microarray analysis.

SC ADIPOSE TISSUE and PLASMA PRIMER SET	
Candidate miRNAs	Exiqon ID
has-miR-15b-5p	002042432
has-miR-532-3p	00204003
has-miR-127-3p	00204048
has-miR-122-5p	00205664
has-miR-6803-5p	02111873
has-miR-3620-5p	02102506
has-miR-16-5p	00205702
has-miR-4529-3p	02118987
has-miR-877-5p	00205626
has-miR-642b-3p	02103759
has-miR-4707-5p	02116591
has-miR-668-5p	02108164
has-miR-425-5p	00204337
has-Let7i-5p	204394
snU6	203907

## Data Availability

Data are available as Appendix A.

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
