# Peer review of "miRNA Expression Profiling in Subcutaneous Adipose Tissue of Monozygotic Twins Discordant for HIV Infection: Validation of Differentially Expressed miRNA and Bioinformatic Analysis"

_ijms, 2022, doi:10.3390/ijms23073486_

Round 1

Reviewer 1 Report

The paper is unique in that it uses twin specimens with identical genetic backgrounds and is likely to interest readers. Moreover, a systematic analysis of miRNA is carried out, and it is another advantage of this manuscript. My comments are as follows,

(1) The specimens were collected from twin and this point is the strength of this manuscript. but it should be clearly indicated whether it is an identical twin or a biovoid twin. This information is very important in considering genome diversity.

(2) The significance of analyzing microRNAs is described in the introduction, but it might be better if more precedent is given about the reason why the microRNA analysis was done, and the necessity is shown.

(3)  Although analysis has been conducted on which gene expression is affected by the altered microRNAs, micro RNAs were chosen according to probable phenotypes indicated by previous studies.  This is a conservative approach, but it may not lead to the discovery of new mechanisms. This limit should be properly described in the discussion.

(4) Although the difficulty of obtaining specimens from twins is understandable, the small sample size may be a problem. It is desirable to add any analysis that can compensate for the small sample size if any.

Reviewer 2 Report

The authors present the expression profiles of two twins, HIV+ vs HIV-, with respect to their miRNA repertoire, to identify biomarkers for HALS independently of a genetic background. The work is easy to read and is well presented. My main concern is the lack of statistical analysis due to the low sample size, and some caution needs to be advised. The authors mention this drawback and tried to compensate by using technical replicates as biological replicates. Another important issue is the lack of data availability, raw data and final results should be open to the readers for further scrutiny and promote transparency.

Some comments:

Line 61: (9)

Line 67: was is

Line 86: <-2 ?

Line 87: Are the 198 miRNAs in HIV- twin from both plasma and SC?

Line 94: heatmaps

Line 143: Not sure what [EB1] refers to

Line 150: “is disadvantaged” sounds odd. Maybe rephrase as: “One disadvantage of …” or “The low sensitivity …”

Line 174: Plasma

Line 431:  Extra 5. At the end of the line?

Line 432: Extra “

Figure 1 legend: is it really values lower than 2 as FC? Isn’t it -2? Maybe specify as you did in the methods

What is the % of miRNA shared between plasma an SC, would be an interesting fact to know, maybe even add a Venn diagram in Fig 1?

Figure 5 Has only B and C panels, I assume Figure A is in Figure 6

Figure 5 legend: A note that the -log10 pvalue is plotted, where the higher the number the more significant.  And maybe mentioning that 1.3 corresponds to 0.05 as reference or even marking it in the plots

Figure 6 legend: HIV+ and HIV- twins

Please indicate version of databases GO and KEGG

The raw data must be publicly available, please deposit in a suitable database (ENA)

microRNA data analysis: were the defaults used in Partek? Or were there any deviations

FC values of all miRNA as Supplementary
